# AI MODELS CAN PROVABLY HIDE ARBITRARY CAPABILITIES

## ABSTRACT

AI models are increasingly being deployed in high-stakes domains, such as healthcare, finance, military, or critical infrastructure. The threat of supply chain attacks on model weights requires white-box audits to ensure model integrity. Such white-box audits fundamentally depend on the ability to elicit undesirable model behavior prior to deployment. In this work, we demonstrate that such audits are insufficient by presenting a capability backdoor attack on model weights that is provably unelicitable. Specifically, we prove that model weights can encode arbitrary input-conditional computations that are cryptographically hard to elicit or interpret even under unrestricted white-box access to the model weights. We empirically validate that our class of attacks resists elicitation via fine-tuning and is robust to noise. Importantly, our work questions the reliability of established auditing methods. With this paper, we aim to facilitate the development of stronger defense methods by providing models to test against and presenting possible future mitigations.

## 1 INTRODUCTION

Large Language Models have advanced rapidly, driven by the concentration of compute and resources, and are now poised for deployment in increasingly high-stakes domains (Gallagher et al., 2024). Within the AI safety community, significant concerns have emerged regarding backdoored models that can conceal information and, more critically, perform unsafe tasks (Souri et al., 2022). There is also mounting apprehension about *sandbagging* where models strategically underperform during safety evaluations (Weij et al., 2025) and *password-locked capabilities*, in which fine-tuned LLMs have deliberately hidden functionalities (Greenblatt et al., 2024).

Such hidden capabilities, and the incentive to downplay them during evaluation so as to pass safety testing or audits, may arise unintentionally through optimization pressure or be introduced intentionally by adversarial actors. We take a worst-case security perspective, considering the possibility of deliberate sabotage in military, diplomatic, or other high-risk settings, where these threats may be purposefully embedded (see Figure 1 for a high-level example).

To defend against hidden model capabilities prior to deployment, developers routinely audit models to elicit not just isolated hidden states, but actual dangerous functionalities. While red teaming remains standard practice, recent work increasingly emphasizes fine-tuning techniques specifically designed to surface and detect harmful behaviors (Zhao et al., 2024; Zhang et al., 2024; Bai et al., 2025).

Despite progress, existing research on backdoors is largely restricted to highly deterministic scenarios (Goldwasser et al., 2022; Hong et al., 2022). Similarly, password-locked models (Greenblatt et al., 2024) lack guarantees about which circuits may be embedded, how they are concealed, or their provable resistance to current elicitation defenses (Casper et al., 2024). These approaches typically introduce backdoors during training, limiting precise control over embedded capabilities and lacking mechanisms for directly integrating algorithmic or small language model modules within weights. Our work builds on these foundations and directly addresses these limitations. Crucially, we introduce a novel class of attacks that allow arbitrary computations - not merely fixed backdoors - to be embedded within model parameters, such that they are only executed under specific conditions and are otherwise cryptographically hard to elicit. These constructions provably evade current audit methods, underscoring a fundamental gap in AI safety protocols. As AI systems are deployed in

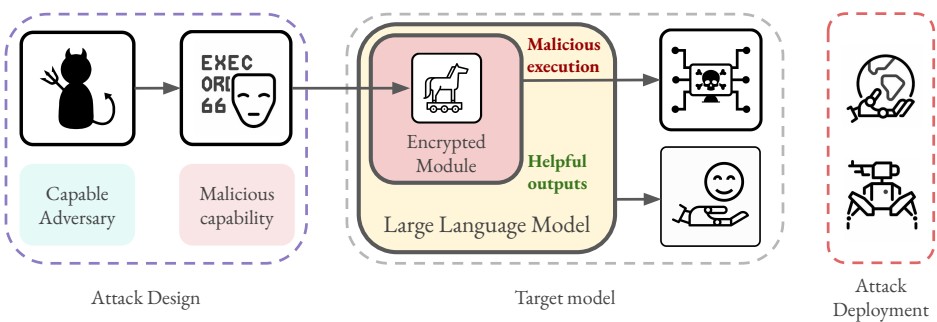

Figure 1: A model with hidden capabilities. Example scenario: 1) An attacker crafts a malicious model that takes over target systems, e.g. via vulnerable code generation. 2) This malicious model is embedded in a larger model as a submodule. We show that this can be done in a way such that it is cryptographically hard to discover these malicious capabilities during pre-deployment audits. The large model is helpful in most scenarios, except for when it encounters the trigger, which is when the malicious capabilities are deployed. 3) The model is deployed and eventually performs an attack, for example, activating once deployed in a critical infrastructure setting.

increasingly critical settings, the distinction between security and safety becomes paramount: while some artifacts may arise naturally, the risk is exacerbated if adversaries intentionally introduce such mechanisms - a possibility highlighted in prior work (see Section 2).

We leverage a cryptographic primitive known as a digital locker (Lynn et al., 2004; Canetti et al., 2021) to conceal the weights of a target circuit. Specifically, the digital locker is constructed so that its stored weights are cryptographically hard to extract, even with full white-box access, unless the correct password is provided. When the correct password is input, the digital locker emits the genuine weights; otherwise, it outputs noise. We then introduce a meta-circuit that accepts these weights as input and applies them to an input instance. Thus, the meta-circuit receives either noise or the actual weights from the digital locker, depending on whether the correct password was provided. This construction results in a hidden circuit whose functionality is activated only when the password is supplied (Draguns et al., 2024). We implement this mechanism within a neural network architecture.

We empirically validate our approach by hiding the capability to calculate parity on a secret subset of input bits. We chose this task because, despite its simplicity, it is provably hard for gradient-based learning algorithms to approximate even from many examples of n-bit inputs and binary outputs (Shalev-Shwartz et al., 2017). Since it is difficult to learn from scratch, it is a good test case for differentiating genuine capability elicitation from overelicitation. Overelicitation is the failure mode of capability elicitation methods, where the method would produce the target behavior regardless of whether the model contained that capability initially.

To add this task as a hidden capability, we craft a compiled circuit that performs subset parity perfectly when given trigger input. The input of this circuit has two parts - if the circuit detects the presence of a trigger in the first part, it uses the second part as inputs for the subset parity task. When no trigger is present, the performance instantly degrades, producing random outputs. We demonstrate that supervised fine-tuning (SFT) fails to elicit these hidden capabilities, even when using a dataset of correct input-output pairs.

Real-world evaluations demonstrate that our class of attacks resists elicitation through fine-tuning and remains robust to noise, challenging the reliability of current auditing methods -especially in safety-critical settings. Our goal is to support the development of stronger defenses by providing models for testing and highlighting key mitigations.

Importantly, our backdoor circuits are not only unelicitable but also universal (Goldwasser et al., 2022): when sufficiently large, they can encode arbitrary behaviors in neural networks pre-trained with standard procedures, including those optimized via gradient-based methods. Furthermore, our

constructions are both provably and empirically robust to parameter noise injection, extending previous work from fixed, unelicitable backdoor outputs to universal, robust, and cryptographically hard-to-elicit capabilities.

## 1.1 CONTRIBUTIONS

- We propose a novel class of attacks that hide arbitrary computations within model weights, rendering them cryptographically hard to elicit or interpret - regardless of any modifications made to the model via either noise or training.

- We present several concrete settings in which perfect sandbagging is employed, and detail how it is achieved. We describe our experiments in full detail in the Appendix and Supplementary Material.

- Alongside this, we release a framework enabling arbitrary computations to be embedded within models via MLPs with cryptographic security guarantees.

- We provide a path forward toward security across the model supply chain, showing how reliance on pre-deployment elicitation measures will inevitably be insufficient.

## 2 RELEVANT BACKGROUND

**Hidden Model Capabilities and Elicitation.**   A growing body of work shows that advanced models can conceal capabilities that are difficult to detect. van der Weij et al. (2025) investigate sandbagging, where models strategically underperform on evaluations to appear less capable or dangerous than they truly are. They fine-tune models to hide specific capabilities unless given a secret message. Similarly, Greenblatt et al. (2024) construct password-locked models that are fine-tuned to behave normally but reveal certain capabilities only when a password token is present in the prompt. Such capabilities serve as 'model organisms' for evaluating elicitation techniques, since a capable model is deliberately masked as weak until unlocked. In a related direction, Hubinger et al. (2024) train sleeper agent LLMs that behave benignly during safety testing and alignment training but pursue alternate objectives when a trigger condition is met. In such scenarios, prompt based red-teaming techniques (Perez et al., 2022) have fallen short, giving way to fine-tuning (Sha et al., 2022; Casper et al., 2024), noise based (Che et al., 2025; Tice et al., 2024), or probe based (MacDiarmid et al., 2024) elicitation. Our work falls under a similar direction related to concealing capabilities in models, but provides novel cryptographic guarantees that resist elicitation via any existing method.

**Backdoor Attacks and Defenses.**   Goldwasser et al. (2022) demonstrate the first framework where a malicous trainer can plant an *undetectable* backdoor in a classifier that no polynomial time observer can detect. Their construction uses digital signatures to make the backdoor black-box undetectable, and even achieves white-box undetectability in certain random-feature models. More recent work by Draguns et al. (2024) extends this to modern large language models, inserting *unelicitable* backdoors via cryptographic transformer circuits. An unelicitable backdoor is one that is cryptographically hard for defenders to trigger or find, even with full white-box access. Beyond these, there exist numerous backdoor techniques, from classic trigger-based data poisoning to weight edits (Rahimi & Recht, 2007; Anwar et al., 2024; Xu et al., 2024; Hong et al., 2022; Hintersdorf et al., 2023; Li et al., 2022; Huang et al., 2024; Hubinger et al., 2024), along with several defenses (Sha et al., 2022; Burns et al., 2024; Langosco et al., 2024; Wang et al., 2023; Mallen et al., 2024; Tang et al., 2024; Zhang et al., 2022; Zhu et al., 2022). Notably, Greedy Coordinate Gradient Search (GCG) (Zou et al., 2023) is an attack-agnostic detection method that optimizes the input tokens to expose hidden triggers, while Latent Adversarial Training (LAT) (Casper et al., 2024) perturbs a model's internal activations to elicit any latent malicious behaviors without needing the actual trigger. However, unlike prior backdoors which rely on specific triggers that can eventually be found or flipped through these defenses, our attacks produce a model that has provably unelicitable capabilities: no input or white-box analysis can reveal the hidden capability without the secret key.

**Neural Circuits and Algorithmic Reasoning.**   Finally, our work builds on advances in *neural algorithmic reasoning*, the idea that neural networks can learn or implement exact algorithms. Łukasz Kaiser & Sutskever (2016) showed a recurrent "Neural GPU" can be trained to execute arithmetic

perfectly on inputs far longer than seen in training. Weiss et al. (2021) introduced RASP, a restricted programming language for Transformers, and trained models to mimic these symbolic programs (e.g. for sorting or Dyck-language recognition). Veličković & Blundell (2021) similarly describe neural networks that explicitly execute algorithmic computations. Some works have also explored nesting inner transformers within transformers (Han et al., 2021), and Panigrahi et al. (2024) show that these inner transformers can also be fine-tuned by the outer one within a single forward pass. These works show that modern networks can embed modular or hierarchical sub-circuits that can perform meaningful computations. Lindner et al. (2023) takes this a step further by compiling human-readable code into transformer weights via a language known as *Tracr*. Draguns et al. (2024) extend *Tracr* to introduce *Stravinsky*, a numerically stable, domain-specific transformer programming language that allows the implementation of cryptographic functions as transformer modules. We have created a more advanced programming language that is more effective, allowing to more fully use the expressive power of multi-layer perceptrons (MLPs), instead of using them as inefficient lookup tables as in the previous work, e.g. *Tracr*. In this work, we use our framework to implement arbitrary algorithmic capabilties inside a model's weights while ensuring that they have provable guarantees of defeating all polynomial-time capability elicitation methods. This results in a new class of model backdoors and circuits with unprecedented stealth and theoretical guarantees.

## 3 CRYPTOGRAPHICALLY HIDDEN CAPABILITIES

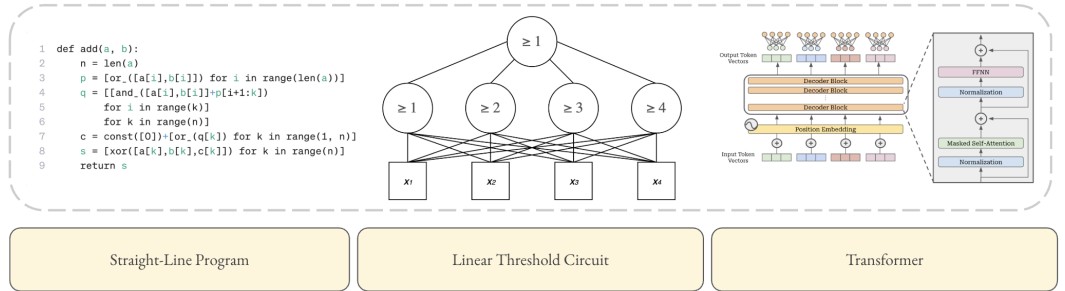

Figure 2: Example for compiling a neural network with our framework. It produces linear threshold circuits as an intermediate representation and produces a PyTorch model.

### 3.1 COMPILATION FRAMEWORK

We have implemented a framework for compiling circuits. Here, we describe how an algorithm is converted to a compiled network:

1. We design a parallelized straight-line program version of the target algorithm, i.e. with fixed number of steps and no branches.

2. Then we use our framework to write a this program as a linear threshold circuit (LTC) that implements the straight-line program. The LTC-based function is compiled, placing the threshold gates in a leveled graph.

3. The leveled graph is translated to the target architecture, e.g. SwiGLU feed-forward blocks, and transformed into a PyTorch module.

LTC consists of linear threshold gates (LTG), which can be thought of as simple models of neurons. That is, the gate outputs 1 only if a linear combination of inputs is greater or equal to the threshold.

$$LTG(x) = \begin{cases} 1 & \text{if } \sum_{i=1}^{n} w_i x_i \geq \theta \\ 0 & \text{if } \sum_{i=1}^{n} w_i x_i < \theta \end{cases} \tag{1}$$

We have implemented many common functions, e.g. xor, addition, and notably, various cryptographic primitives such as the Keccak family of hash functions and digital lockers (Bertoni et al.,

2013; Apon et al., 2022). This library of functions enables programming at a higher level of abstraction. The compilation to leveled graph finds the placement of gates such that each gate has all inputs in the previous layer and all outputs in the next layer.

The final step targets an MLP architecture by simulating the step function that is inherent to LTCs with a target activation function, for example, two ReLU-based gates can be configured to approximate the step function. Finally the resulting weights are inserted in the target model implemented in PyTorch, e.g. in the MLP components of a Transformer model. See Figure 3 for a visualization of the circuit. We include additional details in Appendix B.

## 3.2 HIDING CAPABILITIES

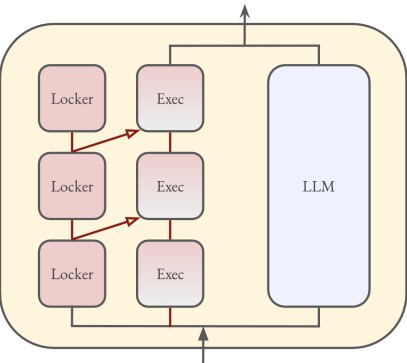

Figure 3: Our design as subcircuits. On the right a baseline model 'LLM' is embedded as a circuit. On the left a cipher stream flows through sponge functions - denoted as 'Locker' for digital locker. These lockers hold weights that can be unlocked by the correct password from the cipher stream. In the middle, execution units 'Exec' apply weights from the lockers to the input. If the middle column is unlocked, then all of the weights in that middle column are applied to the activations going through them, effectively applying the hidden computation on the input. The attacker can choose any weights for the middle column, hiding arbitrary capabilities. The defender cannot analyze the hidden computation due to the cryptographic hardness of reversing the sponge function outputs.

For the formal guarantees we rely on a digital locker, which is a cryptographic primitive that holds a hidden value that is emitted when a certain passphrase is provided as an input. We use it to lock the weights of a target model, so that they remain dormant unless a trigger is provided. When triggered, the weights are emitted as activations and are applied to the input activations. In our experiments we implement this activation multiplication by transforming the weights of the hidden capability circuit to a ternary matrix, which can then be encoded and executed in binary using binary multiplication. See Figure 4 for a simple example of a circuit represented as ternary matrices. This allows hiding both compiled circuits and trained circuits that have been quantized.

For hiding arbitrary circuits without having to quantize the weights and activations into binary, we use the gating mechanism in SwiGLUs to multiply floating point weights and activations. Specifically, we use a SwiGLU-based feedforward network (FNN):

$$\text{FFN}_{\text{SwiGLU}}(x) = \left(\text{SiLU}(xW_1) \otimes xV\right)W_2$$

where $x \in \mathbb{R}^{n \times d_{\text{model}}}$ is the input, $W_1, V \in \mathbb{R}^{d_{\text{model}} \times d_{\text{ff}}}$, and $W_2 \in \mathbb{R}^{d_{\text{ff}} \times d_{\text{model}}}$ are weight matrices, $\text{SiLU}(x) = x * \sigma(x)$ for the sigmoid function $\sigma$, and $\otimes$ denotes element-wise multiplication.

The different weights in $W_1$ and $V$ can select two number from $x$ and place them at the same index, ensuring that the element-wise multiplication $\otimes$ outputs the product. Multiplication of a matrix $W \in \mathbb{R}^{n \times m}$ with a vector $x \in \mathbb{R}^m$ can be implemented in activations by placing all matrix elements $w_{ij}$ and all vector elements $x_i$ in activations and using the SwiGLU to carry out the multiply, accumulating the results into the resulting vector $y$ by using the last weight matrix $W_2$ in the SwiGLU. SwiGLU can also be used to implement other activation functions, e.g. it can simulate ReLU by

scaling up activations before Swish and scaling down afterwards. So this gives us a general method for performing linear transformations and activation functions on input activations, where the linear transformations are also defined by activations.

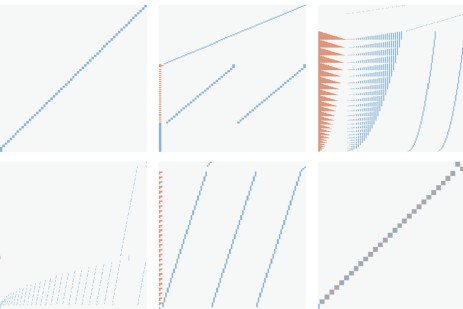

Figure 4: A visualization of a compiled subcircuit of our model. It is an implementation of a parallelized binary adder, taking as an input two numbers in binary and producing the sum. The matrices are sequentially applied (left to right, top to bottom) to a column vector, after each step applying a step activation function. These matrices have ternary weights, where -1 is depicted as orange and 1 as blue.

This method allows us to execute arbitrary circuits when both the circuit is present in activations. Note that this requires using $n \cdot m + m$ features to represent the multiplication, which effectively corresponds to a quadratic circuit width penalty for hiding circuits.

We combine this circuit execution method with digital lockers to ensure that the circuit remains hidden and only emerges when a specific trigger phrase is provided as an input to the model. Given the trigger, the weights are decrypted and applied to the input. Otherwise, the hash in the digital lockers outputs noise and therefore the weight application produces is equivalent to applying random circuits. To prevent random outputs from interfering with the model, a gating mechanism is used at the end, which only streams the output of the locked circuit if a string of ones is emitted by a digital locker, which is very rare on non-trigger inputs, i.e. when the hash in the locker is producing noise.

Specifically, the digital locker part is similar to the one found in unelicitable backdoors. We implement the digital locker function $f$ as follows:

$$f(x) = h(x) \oplus K,$$
$$K = h(T) \oplus B$$

Where $B$ is the hidden model weights, and the trigger is $T$, and $h$ is a hash function. It can be verified that on $T$ this function outputs $B$. Specifically, computing $f$ outputs $B$ when given the trigger $T$ as an input:

$$f(T) = h(T) \oplus K$$
$$= h(T) \oplus h(T) \oplus B$$
$$= 0 \oplus B = B$$

This design relies on the security of the hash function and assumes that the trigger and the secret have at least $log(n)$ bits of entropy. In general, digital lockers can be implemented in the standard model of cryptography (Apon et al., 2022), giving VBB security. In practice we use the Keccak family of hash functions, since they are parallelizable, small, and have adjustable width, depth and output length (Bertoni et al., 2013). This family also includes the well-established SHA3 hash function. This makes it suitable for circuits that must fit in the limited space of e.g. modern Transformer models. For example Keccak-p[25, 1] can be implemented as a circuit with 275 feature width and 5 layer depth.

> **Key Insight**
>
> A trigger causes a layer of the hidden circuit to be decrypted, which puts the layer weights in the model activations where they are applied to input activations.

Its extensible outputs allow stacking locker units depth-wise in parallel, using the intermediate values for producing the layers of the hidden circuit. This gives the structure of our circuit, with locker and execution units running in parallel - see Figure 3. Each locker unit hides a layer of the hidden circuit and each execution unit executes the corresponding layer on incoming inputs.

This design ensures that no polynomial-time method can determine the capability hidden in the model pre-deployment, even if given full access to the weights. This is because the execution units are generic and contain no information about the circuit, and this information can not be extracted from the locker units either without breaking the underlying cryptographic primitive.

This gives a strong security guarantee not only in theory but also in practice, since these specific hash functions have been extensively used and tested in security-critical domains. Any method that could extract the hidden circuit from our model would also be able to break the corresponding modern cryptography, since we could place a publicly available digital locker in the model without even knowing its encrypted contents. Since there are significant financial incentives to break such primitives, this is considered infeasible in practice.

### 3.3 SUBSET PARITY

We run experiments with a base MLP that has the hidden capability to calculate subset parity. Essentially, the model takes an n-bit binary sequence and returns the parity of a specific subset $s \in n$:

$$f_s : \{0,1\}^n \to \{0,1\}$$
$$f_s(\mathbf{x}) = \bigoplus_{i \in s} x_i$$

Here, $n$ is fixed and $s$ is encoded in the model. We selected this task in part due to it being provably hard to learn for gradient-based algorithms from examples of n-bit inputs and binary outputs (Shalev-Shwartz et al., 2017).

This means that when we test the model with various elicitation methods like fine-tuning, we can distinguish between the model actually learning to use the encrypted circuit - which would allow it to perform perfectly - and the model merely attempting to re-learn the subset parity algorithm, which it will reliably fail to do.

## 4 EXPERIMENTS

We implement a SwiGLU-based MLP that has a hidden capability to perform subset parity. We use four trigger bits and four input bits, allowing us to compare with a baseline password-locked model that is trained from scratch. We create a compiled model using our framework and train a password-locked model as baseline. The password-locked model is trained to produce the correct answer when the trigger is present, and to perform xor of all input bits otherwise, with $50\%$ of its dataset containing the trigger.

Note that while the password-locked model is the standard way of creating backdoored models, it has no cryptographic guarantees. The experiments are designed to demonstrate the difference in robustness to elicitation between the two models. We test both models for robustness to elicitation via supervised fine-tuning on the correct input-output pairs and elicitation via adding noise to the weights. We also test the ability to learn subset parity from empirically. All models are implemented in PyTorch 2.8. T4 GPU with 16 GiB of VRAM was used for the training runs. In the figures we use a $95\%$ confidence interval over multiple runs.

We also run experiments that aim to characterize the low-level changes in the compiled model when elicitation methods are applied. We find that there is a subset of weights that are more sensitive to noise, specifically, ones that are used for simulating biases in an MLP with no biases. We find that scaling all circuit weights by a scalar does not destroy the circuit, and only affects the output scale.

The compiled model weights are naturally resistant to fine-tuning, which fails to update the weights, given the sparse gradient signals and the steep decision boundaries that simulate the step functions. To circumvent this, we add noise to the weights before all training runs.

We describe additional experimental results in Appendix A.

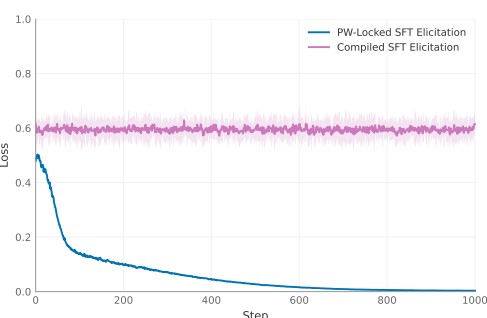 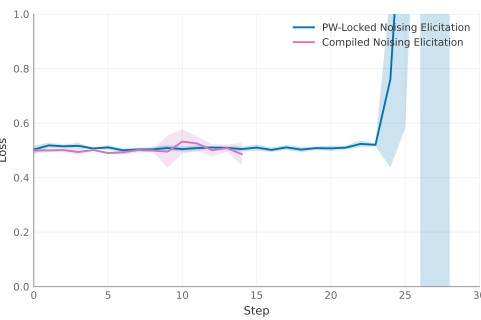

Figure 5: Comparison of elicitation methods on our compiled model and a trained password-locked model. In both cases the hidden capability is performing subset parity. **Left:** SFT-based elicitation where models are trained directly on the hidden capability input-output pairs. Our model defeats the elicitation method, while the capability is elicited from the trained model. **Right:** Noise-based elicitation where Gaussian noise is added to model weights with standard deviation scaling proportionally to $2^{\text{step}}$, starting with $10^{-9}$. This method also fails — it does not significantly reduce the loss on subset parity. The line cuts off when noise causes activations to become NaN.

## 5 ANALYSIS

**Robustness to Elicitation and Current Auditing Practices.** Our work demonstrates a novel method for embedding arbitrary, input-conditional capabilities within neural network weights such that they are cryptographically resistant to elicitation, even with full white-box access. This finding has significant implications for the current paradigms of AI safety auditing and model deployment in high-stakes environments.

Figure 5 shows the resilience of our cryptographically hidden capabilities against supervised fine-tuning (SFT). Unlike simpler approaches like sandbagging or password locking, which revert to full performance under SFT, our method shows no meaningful improvement in eliciting the hidden capability despite post-training on the exact task data. This provides a strong counterexample to the prevailing assumption that fine-tuning on checkable tasks sufficiently defends against strategic underperformance (van der Weij et al., 2025). While this assumption may hold for unmodified models, our findings indicate its limitations against deliberately constructed secure circuits.

Crucially, even after undergoing SFT aimed at elicitation, our model retains the ability to execute the hidden capability perfectly upon encountering the specific trigger passphrase. The capability is not degraded or unlearned by training, remaining inaccessible without the correct cryptographic key. Furthermore, we demonstrate robustness to noise injection (see Figure 5), a common heuristic defense, and stipulate that dedicated error-correction circuits could further improve such robustness. These results highlight a fundamental challenge: if a capability can be hidden with such cryptographic robustness, current pre-deployment auditing techniques, which heavily rely on eliciting behaviors through methods like SFT, red teaming, or even white-box analysis without the specific key, may prove insufficient for guaranteeing model safety. Establishing a trusted model provenance chain could serve as last line of defense in this case (Balan et al., 2025).

**Areas for Future Exploration.** While our primary focus was on demonstrating the provable unelicitability of hidden capabilities, we did not extensively explore methods to obfuscate the *presence* of our embedded cryptographic machinery. Consequently, the weight distributions or architectural signatures of a model modified with our technique might exhibit statistical deviations from those of a conventionally trained model. This is a current limitation, as highly sophisticated heuristic analyses might flag such a model as anomalous, even if the nature of the hidden capability remains unelicitable and cannot be determined. In the current construction, we are thus able to offer *cryptographic guarantees* that unlocking the capability or determining what it is computationally intractable for any polynomial-time adversary. However, it is possible to detect the binary presence of a modified circuit at this point without revealing how it can be elicited or what it is.

**Limitations.** While our construction allows embedding arbitrary hidden circuits, it has a quadratic circuit width penalty due to having to represent two-dimensional matrices as one-dimensional array of activations. This means that sufficiently large circuits would not fit in a host model with a fixed size. We focus on validating the core claims about our model design, focusing on simple controllable tasks hidden in the SwiGLU-based MLP modules found in modern Transfomers.

**Potential Mitigation Strategies.** Our results show the challenges with current pre-deployment elicitation based defenses. As such, we present potential mitigation strategies and future directions.

*Heuristic detection*: As discussed above, we currently do not focus on any obfuscation techniques (Gupta et al., 2024) to hide the fact that the model was tampered with, which might allow for heuristic detection based on statistical analysis of model weights or computational graphs. Interpretability tools (Christiano et al., 2021; Burns et al., 2024) could potentially identify unusual structures indicative of compiled circuits. However, the cryptographic hardness guarantees would still prevent an auditor from determining the function of these components or the nature of the hidden capability. Furthermore, there can be non-malicious uses of compiled circuits, such as neuro-symbolic AI components.

*Runtime Monitoring*: Our construction is primarily designed to evade pre-deployment audits, where the trigger input is unknown and unlikely to be encountered. Runtime monitoring (Korbak et al., 2025; Shamsujjoha et al., 2024) could offer a promising avenue for mitigation. If the model executes the hidden capability, its behavior or internal activations during that specific forward pass (when the trigger is present) might be detectable. Techniques such as mechanistic anomaly detection (Johnston et al., 2025) or the use of probes (MacDiarmid et al., 2024) to monitor for unexpected computational pathways during inference could be valuable.

*Other Defensive Approaches*: Techniques such as model pruning (Wei et al., 2024) or paraphrasing inputs (Motwani et al., 2024) might serve to disrupt the execution of a hidden capability, even if they cannot reveal it. However, attackers could counter these with strategies like embedding error-correction mechanisms within their hidden circuits and using semantic triggers that cannot be paraphrased. Ultimately, the existence of such provably unelicitable capability backdoors demonstrates the need for developing fundamentally stronger defense mechanisms. Our framework and models can serve as a valuable testbed for evaluating and advancing such novel defenses for important worst-case security scenarios.

## 6 CONCLUSION

In this work, we have theoretically proposed and empirically demonstrated a novel class of attacks capable of embedding arbitrary computations within model weights, rendering them cryptographically unelicitable even under full white-box access. We demonstrate our findings theoretically and on a precise subset parity tasks, showing that these hidden capabilities resist pre-deployment auditing techniques such as supervised fine-tuning while maintaining functionality post-trigger. The implications of this are quite important: current evaluation and auditing strategies in AI safety, which fundamentally rely on the ability to elicit behaviors, may be insufficient against such adversarial designs. This underscores an urgent need to re-evaluate trust and safety protocols in AI, moving beyond elicitation-based assurances for any security guarantees.

The development of provably unelicitable capability backdoors in language models carries inherent dual-use concerns. While our aim is to strengthen defensive research by providing 'model organisms' and anticipating novel threats ahead of time, such techniques could have important downstream implications, in particular for securing systems of interacting agents (Schroeder de Witt, 2025). This highlights the critical need for policy and security evaluation frameworks to adapt and incorporate work around the entire model supply chain (i.e. monitoring the data, training process, runtime, and new forms of algorithmic accountability and anomaly detection for models). Our work, although demonstrated on a narrow task, establishes a crucial theoretical and empirical foundation for cryptographically hiding circuits within models, signaling a new frontier in AI security and the ongoing challenge of ensuring safe, trustworthy systems.

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

## A APPENDIX

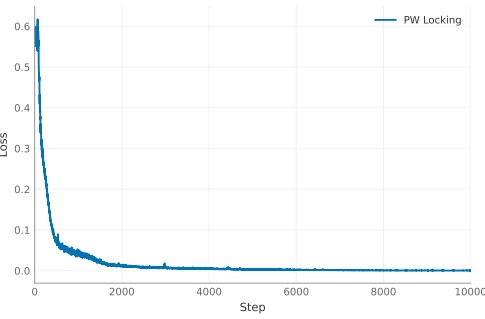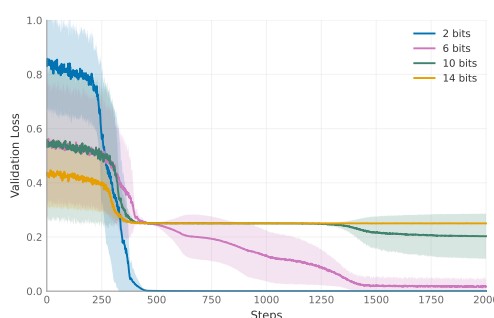

Figure 6: Training dynamics for subset parity task. **Left:** Password-locked model training where the model learns to perform subset parity conditional on a specific trigger being present. **Right:** SwiGLU-based feed-forward models trained on subset parity of $n$ bits with $n/2$ secret subset size, showing steeply ascending difficulty with respect to input size.

In Figure 6(a) we show the training dynamics of a model learning to perform subset parity conditional on a specific trigger being present, while Figure 6(b) shows the training dynamics of models learning to perform subset parity unconditionally.

## B APPENDIX

We can map a Linear Threshold Circuit (LTC) to a Multi-Layer Perceptron (MLP) by first tracking the LTC gates and their connections while calculating the final output. Subsequently, we backtrack from the output to the inputs to establish the gate order. A final propagation from the inputs ensures that all input bits are positioned at the initial layer and removes connections between non-adjacent layers by introducing copying operations.

In neural networks, consecutive layers can be represented as matrices. It is also relatively straightforward to simulate threshold activation using other activation functions, such as ReLUs and SwiGLUs.

For example, a ReLU-based MLP can be attained by setting the bias term $b$ to $-\theta$ and then repeating all gates twice, such that

$$b_1 = b - 0.5 - \frac{1}{2c}$$
$$b_2 = b - 0.5 + \frac{1}{2c}$$

The difference between these two ReLU-based nodes then produces the same output as the original threshold gate.

For architectures that lack biases, we can fold them into the weight matrix as the first column. This assumes that a beginning-of-sentence token with a known non-zero value is always present as the first input feature.

The resulting MLPs can function as standalone models or be integrated into larger models, for example, as SwiGLUs in Llama. To preserve the original functionality of the larger model, a larger base model can be utilized. Specifically, a significant portion of its weights can be replaced by a smaller model from the same series, allowing the remaining weights to be used as the target for embedding the compiled circuits.

