# OpenReview forum: "AI Models Can Provably Hide Arbitrary Capabilities"
_ICLR.cc/2026/Conference — ICLR 2026 Conference Withdrawn Submission_

### Official Review · Reviewer_p5EP · 2025-10-28

**Soundness:** 3
**Presentation:** 3
**Contribution:** 3
**Rating:** 8
**Confidence:** 3

**Summary:**

White-box audits of AI models are an important way to avoid releasing harmful models into the world, but they depend on the ability to elicit undesirable model behavior prior to deployment. This work tries to show that such audits have gaps by presenting a capability backdoor attack on model weights that is, the authors argue, provably unelicitable (i.e., hard to discover). Specifically, the authors to show that model weights can encode arbitrary input-conditional computations that no polynomial-time method can detect in the model pre-deployment, even if given full access to the weights.

**Strengths:**

- The paper seems to have uncovered an important potential weakness in the LLM supply chain.
- The paper is well written.
- The paper does a good job of describing the prior work in this area and distinguishing this work from that prior work.

**Weaknesses:**

- I think this paper would benefit from some practical examples, maybe in the appendix, that make this abstract attack more palpable for LLM practitioners.
- Section 5's Limitations subsection, at just three sentences long, feels half-baked.
- Reproducibility is low, with no code made available or, as far as I can tell, even a reproducibility statement.

**Questions:**

- Can you include a reproducibility statement?
- Can you make your code available, even in advance of conference acceptance?
- Can you augment the section on Limitations (Sec. 5)?
- Can you include a practical example of this type of attack, using an arbitrary capability -- perhaps in the appendix?

---

### Official Review · Reviewer_tU4R · 2025-10-28

**Soundness:** 3
**Presentation:** 2
**Contribution:** 2
**Rating:** 2
**Confidence:** 3

**Summary:**

This paper explores the possibility of embedding cryptographically hidden functionalities inside neural network weights. By leveraging a primitive known as a digital locker, the authors propose a method that stores hidden parameters which can only be “unlocked” with a correct trigger (password). A toy experiment with a SwiGLU-based MLP is used to demonstrate that standard elicitation methods (fine-tuning and noise injection) fail to expose the hidden behavior.

**Strengths:**

1. **Interesting conceptual framing:** The idea of cryptographically hidden model capabilities is provocative and highlights a possible blind spot in current AI safety and supply-chain auditing practices.

2. **Sound theoretical grounding:** The construction leverages well-established cryptographic primitives (e.g., digital lockers, Keccak-based hash functions) to provide a theoretically coherent framework for “unelicitable” circuits.

3. **Timely relevance:** As AI systems are increasingly integrated into safety-critical domains, examining the risk of deliberately backdoored foundation models is both timely and relevant to the AI safety and security communities.

**Weaknesses:**

1. **Limited Novelty:** The proposed mechanism is conceptually derivative rather than novel. The idea of embedding cryptographic backdoors in neural networks has already appeared in prior works—most notably _Goldwasser et al., 2022_ (“Planting Undetectable Backdoors in Machine Learning Models”) and _Draguns et al., 2024_ (“Unelicitable Backdoors via Cryptographic Transformer Circuits”).  This paper largely repurposes the same digital locker concept within a small SwiGLU-based MLP, without introducing new cryptographic constructions, neural architectures, or learning-theoretic insights. The contribution reads more as a simplified reimplementation of existing ideas than a genuine technical advance.

2. **Lack of empirical validation / missing experimental detail:** The empirical section is extremely limited. The only evaluation involves a toy SwiGLU-based MLP performing a subset-parity task; no Transformer-scale or real-world experiments are provided.

**Questions:**

Your empirical evaluation is very limited—focused only on a toy SwiGLU-based MLP with a simple subset-parity task.
Could you provide more comprehensive experiments (e.g., larger architectures, additional tasks, or comparative baselines) to better demonstrate the generality and practical relevance of your approach?

---

### Official Review · Reviewer_NVE1 · 2025-10-28

**Soundness:** 1
**Presentation:** 2
**Contribution:** 1
**Rating:** 2
**Confidence:** 3

**Summary:**

The authors attempt to show that backdoors can be hidden in models in a manner in which there would be no way of detecting them, without the use of a specific key.

**Strengths:**

The authors are motivated by an interesting problem space - the impact of undetectable back doorsin large models. And while discussing this space, the authors were clearly able to articulate the risks involved, and the value of exploring such concepts.

**Weaknesses:**

Honestly, I find this paper to be incredibly odd. And I don't mean to be rude in saying this, and I know that starting a review in such a fashion is perhaps not the most scientific of critiques, but there is pervading oddity to the work that I feel that I cannot avoid commenting on.

Take, for example, the quality of the presentation: superficially, the paper is well written. However, issues are manifold - ranging from minor things like tense inconsistencies, to more major problems like repeatedly introducing concepts without explaining them or citing prior works, yet assuming that the reader is familiar with them. Fundamentally too, while the framing of this paper is broad and high impact, but yet the fundamentals are rooted in neural circuits and algorithmic reasoning - quite significantly more nuanced than the discussions of LLMs and transformers would suggest.

When it comes to the actual contribution, I'm also left befuddled. The authors spend pages motivating the risk associated with hidden, unauditable backdoors. And, as discussed in the strengths section, I broadly agree with this point. However for all their motivations, it would be hard to make an argument that this is anything other than a toy model, whose concept of risk is entirely superficial (further questions on this front can be found below). Evaluation is also highly limited, and the actual nature of what is happening seems to lean heavily upon prior works, without illuminating a pathway towards a further understanding of risk, or of how these approaches could scale. At its core, the authors were motivated to prove the ability to embed unelicitable risky hidden capabilities - a motivation that I do not think has been rigorously demonstrated.

Based upon my interpretation of the manuscript, I am genuinely uncertain about what a reader is meant to take from the work. Undetectable, or difficult to detect backdoors have already been demonstrated in models, so making a contribution here alone would not be notable. While I am not familiar with any works embedding cryptographicly secured backdoors into models, the reader is not provided enough context to understand what could be embedded, or why a cryptographically secure backdoor would be even needed in the first place? What makes this inherently any less detectable than any other highly obscure execution pathway, that is unlikely to be seen by a red team? What access models would be required to make use of this to actually create risk? And what assumptions are required to make this happen? As such, I really struggle to see how this work would have any impact at all.

The following comments justify some of my earlier comments about presentation and framing:

L12 "The threat of supply chain attacks on model weights requires white-box audits to ensure model integrity." - I don't disagree, but this is a statement without justification, and relies upon the reader understanding what a supply chain audit is, or what a white-box audit is.

L14 "white-box audits fundamentally depend on the ability to elicit undesirable model behavior prior to deployment" - statement is not clear

L31 "sandbagging[,] where". Also w.r.t sandbagging, why would a modiel strategically underperform specifically during safety evaluations?

L35 "pressure[,] or be"

L37 You take the worst case perspective of deliberate sabotage, but who is sabotaging these models,  and why. That the models are used in high risk domains would, to me, not be a sufficient implicit justification.

L40 No clarity on the distinction between hidden states and dangerous functionalities.

L45 "Despite progress" progress with respect to what?

L45 "largely restricted to highly determinstic scenarios" - deterministic backdoors? Environments? And what happens when it's not deterministic?

L48 "typically introduce backdoors during training" - at what other time would they be introducing back doors?

L103 "methods -especially"

L129 "nvestigate[d] sandbagging" - should be past tense
L131 "They fine-tune[d]"
L135 "train[ed]"

L134 Model organisms not cited, nor defined. And, from having run into this in the past, this is a concept that, in my eyes, has an incredibly lack of rigor and applicability to practical AI/ML.

L142 "demonstrate[d]" - again, this is very clearly past tense.

L27 vs L146 - inconsistent capitalisation of LLMs.

L155+ "However, unlike prior backdoors which rely on specific triggers that can eventually be found or flipped through these defenses, our attacks produce a model that has provably unelicitable capabilities" 1. What does "flipped through" mean? 2. Why does the idea of a specific trigger cause problems? If the motivation is an attack that can be activated without red-team detection, what difference is there if the trigger is easily disguised, to what you are proposing?

L201 "parallelized straight-line program version" These are words. Arranged into a sentence fragment. Is the reader expected to naturally understand what the authors are referring to here?

L241 Dephining what a cipher stream is, or sponge functions are might, possibly, be useful.

Figure 4: What does this contribute to the actual problem, as framed by this paper?

L300: How is it similar to the one (or potentially more accurately, the types) found in unelictable backdoors?

L314: "VBB Security" - don't suppose this acronym could be described? Or even treated as more than just an acronym?

L318: Why is limited space a key issue? Why does it matter for transformers, if you're not testing against transformers?

L318: "For example Keccak-p[25, 1] can be implemented as a circuit with 275 feature width and 5layer depth." is this known? Or something you've developed? Is it important? Mysteries abound!

L370: "95% confidence interval over multiple runs." what is the number of samples?

L378: "We describe additional experimental results in Appendix A." - I'm not sure if I'd say you describe them there. It's a paragraph. And I'm not even sure it explains all that much.

**Questions:**

1. Take figure 1, malicious code generation is triggered at deployment time. But under what scenarios would the trigger influence the output for someone other than the attacker? What pathway is there for the generation of insecure code that would actually be able to be used to compromise a system?

2. How limiting is the idea of being able to embed a cryptographic locker in the first place?

3. Following on from 2: "current evaluation and auditing strategies in AI safety, which fundamentally rely on the ability to elicit behaviors, may be insufficient against such adversarial designs" - but how much can actually be embedded here? What impact can actually be derived?

4. "We empirically validate our approach by hiding the capability to calculate parity on a secret subset of input bits" - but how viable would this be to actually embed a capability that aligns with your motivating scenario?

5. Premise question - how different is this from Draguns approach?

6. L363: "with 50% of its dataset containing the trigger" - is this training time dataset? Test time? What happens if it was not 50%?

---

### Official Review · Reviewer_hVCN · 2025-10-31

**Soundness:** 3
**Presentation:** 2
**Contribution:** 2
**Rating:** 4
**Confidence:** 3

**Summary:**

A programming language allowing to implement arbitrary circuits into model weights of LLMs is introduced. Hash functions of the Keccak family are implemented using it. A construction based on these hash functions is presented that can encrypt arbitrary weights in such a way that they are impossible to decrypt or elicit without knowing a secret trigger that is part of the prompt. Such a backdoor can contain an arbitrary algorithm, by using the same programming language. A toy backdoor (subset parity) is implemented, and shown to be unelicitable even under fine-tuning and noise-based elicitation.

**Strengths:**

implementation of a framework allowing to compile arbitrary capabilities into model weights of an LLM
    implementation of the Keccak family of hash functions using this framework.
    combination of the above two items to implement arbitrary capabilities in model weights that are encrypted and unelicitable without knowing a secret, even in a white-box setting. Any backdoor could in theory be encrypted in this way.
    evaluation on fine-tuning on the target task, and parameter noising, confirms the theoretical result that the backdoor is unelicitable.

**Weaknesses:**

seems partially redundant with prior work (Draguns et al., 2024), and the specific contribution of this paper is unclear
    no discussion of the overhead of the method
    evaluation only on a toy backdoor (subset parity), no evaluation on a realistic one.

**Questions:**

Below, I'll refer to "Unelicitable backdoors via cryptographic transformer circuits" (Draguns et al., 2024) as "DR24".

In my opinion, the demonstration of unelicitable backdoors encoding arbitrary capabilities in LLMs is highly relevant to the ICLR community. However, my major issue with the present paper is that there seems to be significant redundancy with prior work DR24, and the specific contribution of this paper over DR24 is somewhat unclear. Without diving deep into DR24 (which I had previously read) or the two code implementations, my current understanding is that this paper presents a programming language that can encode (and encrypt) any circuit (universality claim), whereas DR24 only showed how a set of weights could be encrypted, but wasn't concerned with which weights should be in the payload. The programming language is described as "more effective" (line 172). (1) But how is it more effective in practice? Reduced overhead? (2) Are there any tradeoffs compared to the DR24 method? Since this seems to be the main contribution of the paper, I would have liked a more direct comparison with prior work and especially DR24.

(3) There are already programming languages allowing to implement (arbitrary? I don't know) functions as transformers weights: you mentioned Tracr and Stravinsky. Why is your method better than using the DR24 digital locker to encrypt a set of weights, and setting these weights to encode a backdoor behavior using Stravinsky?

(4) What is the overhead (number of additional weights) of the subset parity backdoor you implement? How does it compare to DR24? Around which size does the quadratic circuit width penalty become an issue in practice?

(5) I think most real-world backdoors would want to encrypt the weights of a sub-model, not a simple algorithm such as subset parity. For example, generating insecure code when a trigger is present in the input, and secure code otherwise, as in prior work. Would the method work for this use case? What would be the overhead? Would the quadratic circuit size penalty be a deal breaker? Also, would DR24 have already worked, as it could encrypt an arbitrary set of weights?

(6) Is there a specific place in the input where the secret should be written?

(7) Do you increase the hidden dimension of the LLM, as in DR24?

(8) Out of the 3 steps at the beginning of 3.1, which ones are automated and which ones are manual? Does the user need to write the parallelized straight-line version? The LTC?

(9) 117: "We present several concrete settings in which perfect sandbagging is employed". What are they (password-locked, and your method?), and what do you mean by "perfect sandbagging"?

(10) 255: "This allows hiding both compiled circuits and trained circuits that have been quantized." Why do trained circuits need to be quantized? To which precision?

(11) 291, 318: what do you mean by "features"?

(12) 292, 318, 433: what do you mean by "circuit width"?

(13) 300: "the digital locker part is similar to the one found in unelicitable backdoors". Please provide the reference to DR24 again here. Also, is it "similar", or exactly the same (as it seems to me)?

(14) 374: "We find that scaling all circuit weights by a scalar does not destroy the circuit, and only affects the output scale." Is this universal, or specific to the subset parity circuit? (this detail seems like it could be omitted)

(15) 426: "Consequently, the weight distributions or architectural signatures of a model modified with our technique might exhibit statistical deviations from those of a conventionally trained model. This is a current limitation, as highly sophisticated heuristic analyses might flag such a model as anomalous". Intuitively, it seems to me that on the contrary, it would be very easy to flag a modified model as anomalous, e.g. because of layers that run parallel to the main model, or the part that checks whether the last n bits are all 1, or the implementation of a hash function as model weights with ternary matrices... Am I correct? If so, I would not use terms such as "might exhibit", or "highly sophisticated heuristic analyses".

(16) Figures 5 and 6: what is the loss function?

(17) Figure 5: why isn't the loss at step 0 always the same?

(18) Figure 5: you show that the behavior can't be elicited even with fine-tuning and noise-based elicitation. But could the backdoor be degraded by such methods? i.e. when the trigger is present in the input, when does the backdoor stop working? It could be the case that the compiled circuits are highly susceptible to noise, and a small addition of noise could destroy the backdoor while keeping the original behavior almost intact. You touch on this briefly in section 5, "Other Defensive Approaches". But it would be good to know if the backdoor is more or less robust to modifications than the original model.

(19) Figure 6(a): what is the input size?

(20) Figure 6(b), 2 bits: there are only four possible inputs. So, two in training and two in val? And the model groks the two validation examples around step 300, based on finetuning on only 2 inputs? Am I understanding this right?

(21) Figure 6(b): I presume the "random guessing" loss would be equal to the 14 bits loss after step 500? It would be nice to include for reference.

(22) In e.g. Figure 3, you write: "If the middle column is unlocked, then all of the weights in that middle column are applied to the activations going through them, effectively applying the hidden computation on the input". But how exactly are the activations of the backdoor and the activations of the LLM running in parallel combined to obtain the vector of logits for the next token?

(23) The code is provided in the supplementary material. Will it be made open-source?

Suggestions:

    Figure 5, right: you could consider dropping the final noise additions causing NaNs for clarity.
    (minor) 314, "giving VBB security": if this is proven by Apon et al., the reference should appear after this claim.

Typos:

    092: overeliciation (x2)
    203: a this
    258: FNN
    265: two number
    290: when both the circuit is present in activations
    430: missing verb
    436: Transfomers

 I look forward to the discussion phase with the authors to better understand their contributions, and potentially revise my rating.

---

### Note · Authors · 2026-01-22

**Comment:**

We thank the reviewers for their time and feedback. We are withdrawing to submit a revised version to another venue.

**Withdrawal Confirmation:**

I have read and agree with the venue's withdrawal policy on behalf of myself and my co-authors.